# Travel time prediction of urban public transportation based on detection of single routes

**Xinhuan Zhang** [1]*, **Les Lauber**[2], **Hongjie Liu**[3]*, **Junqing Shi**[1‡], **Meili Xie**[1‡], **Yuran Pan**[1‡]

**1** The Institute of Road and Traffic Engineering, Zhejiang Normal University, Jinhua, Zhejiang Province, China, **2** Kansas Public Employee Retirement System, Topeka, Kansas, United States of America, **3** School of Electronic and Information Engineering, Xi'an Jiao Tong University, Xi'an, Shanxi Province, China

☯ These authors contributed equally to this work.
‡ JS, MX and YP also contributed equally to this work.
* zxh@zjnu.cn (XZ); hj_popel@stu.xjtu.edu (HL)

**Data Availability Statement:** All relevant data are within the paper and its Supporting Information files.

**Funding:** This research was supported by Zhejiang Provincial Natural Science Foundation of China (General program) ( Grant NO. LY18G010009),

## Abstract

Improving travel time prediction for public transit effectively enhances service reliability, optimizes travel structure, and alleviates traffic problems. Its greater time-variance and uncertainty make predictions for short travel times (≤35min) more subject to be influenced by random factors. It requires higher precision and is more complicated than long-term predictions. Effectively extracting and mining real-time, accurate, reliable, and low-cost multi-source data such as GPS, AFC, and IC can provide data support for travel time prediction. Kalman filter model has high accuracy in one-step prediction and can be used to calculate a large amount of data. This paper adopts the Kalman filter as a travel time prediction model for a single bus based on single-line detection: including the travel time prediction model of route (RTM) and the stop dwell time prediction model (DTM); the evaluation criteria and indexes of the models are given. The error analysis of the prediction results is carried out based on AVL data by case study. Results show that under the precondition of multi-source data, the public transportation prediction model can meet the accuracy requirement for travel time prediction and the prediction effect of the whole route is superior to that of the route segment between stops.

## I. Introduction

Public transit travel time prediction is one effective measure for improving service reliability, optimizing travel structure, and alleviating traffic problems. Accurate real-time travel time information benefits include reducing passenger waiting time, relieving passenger anxiety, reasonably distributing travel time, improving convenience in taking or transferring busses, enriching public transit services types, enhancing public transit's image and improving its desirability, and providing a rational basis for scheduling [1]. In contrast, conventional transit is a complex non-linear system with multiple parameters, creating a high degree of uncertainty. This complexity makes it challenging to predict travel times, especially when short-

Zhejiang Provincial Educational Committee(Grant NO.Y201738488), the Scientific Research Foundation for the Returned Scholars, Ministry of Education of China(Grant NO.ZC304012027).The funders had no role in study design, data collection and analysis, decision to publish, or preparation of the manuscript.

**Competing interests:** The authors have declared that no competing interests exist.

term travel times are more susceptible to random factors and uncertainties than long-term forecasts. Moreover, as travel time increases, so does the probability of unexpected accidents. The requirement for short-term travel time prediction precision ($\leq$35min) is higher than for long-term, making its prediction also more complex.

Traditional travel time prediction methods based on statistical analysis or mathematical modeling are deficient in intelligence and have weak adaptability. Current methods depend on real-time detection of traffic conditions regardless of precision or accuracy. Some methods based on new mathematical tools, such as neural networks, have a good application effect for some specific problems but lack generalizability. Other knowledge discovery methods embody knowledge-based requirements but have not formed a methodology suitable for traffic data analysis. Therefore, in the big data environment, it is necessary to study travel time prediction technology to more fully meet the data requirements in the operation analysis process of urban public transit and improve its applicability.

This paper studies the travel time prediction of urban public transit based on the Kalman filter in a big data environment. Based on leading research and application in intelligent transit technologies, the theories and methods of current travel time prediction of public transit based on data mining are enriched and complemented. Additionally, a travel time prediction system of conventional transit under the influence of random factors is constructed, and a short-term ($\leq$35min) travel time prediction model with strong applicability is built. These will work congruously to improve prediction's real-time accuracy and adaptability and reduce costs associated with data acquisition. This paper provides a scientific theoretical basis and decision support for the practical work of using intelligent technology to improve the prediction accuracy of travel time. It holds practical application value for four stakeholder groups:

1. Travelers: As a general travel time prediction system of public transit, it can predict travel time information within a certain period. This improves an urban public transit system's service levels and provides passengers with real-time travel time information through multiple channels to facilitate their travel choices.

2. Public transit operators: Accurate travel time prediction is the basis of dynamic scheduling. Using artificial intelligence to perceive and predict the system's status continuously improves travel time predictions in real-time. Establishing a new generation of intelligent information service systems that bases predictions on big data analysis improves and enhances the intelligent service level of public transit. Innovative big data analysis applied to public transit realizes gains in the efficient operation and management of public transportation systems.

3. Public transit managers: Building a flexible and universal public transit travel time model based on data mining in the big data environment can provide public transit managers with decision-support information related to management planning.

4. Public transit policymakers: Strategic public policymakers and planners can use this approach for improving passenger satisfaction, optimizing scheduling schemes, increasing public transit system reliability, making public transit options more attractive, promoting reasonable travel plan creation, and relieving the pressure of urban traffic. These benefits will facilitate the strategic development of public transit.

## II. Literature review

Prior researchers have developed methods to predict travel time [2–4]. This section is an overview of methods for predicting travel time published in the last five years. These methods can

be divided into five categories: Global Positioning System (GPS) based, Neural Network based, Support Vector Machines (SVM) based, Particle Filtering (PF) based, and Kalman Filter (KF) based.

## A. Global positioning system

GPS signal positioning is the most direct method for travel time prediction. Based on bus riders' smartphone Wi-Fi information, Liu et al. [5] presented a model to track and predict the arrival time of a city bus. Automatic Vehicle Location (AVL) and smartphone location can also predict bus arrival time. Farooq et al. [6] presented a prediction system relying on real-time AVL. While using technological solutions such as GPS and AVL, those methods could not use historical information and ignored space features.

Chen EH et al. [7] proposed a generic framework to analyze short-term passenger flow, considering passenger flow's dynamic volatility and nonlinearity during special events. Four different generalized autoregressive conditional heteroscedasticity models and the ARIMA model were used to model the mean and volatility of passenger flow based on the transit smart card data(Contains GPS information) from Nanjing, China. The proposed framework could effectively capture the mean and volatility of passenger flow and outperform the traditional methods in terms of accuracy and reliability. Zhang B et al. [8] focuses on identifying the distribution of regions with high travel intensity and the correlation between travel intensity and points of interest (POIs), based on the online car-hailing data collected in Chengdu, China (Contains GPS information). Zhang HL et al. [9] propose a spatial-temporal generative adversarial network (ST-GAN) to assign the generative factors of traffic flow to the feature vector in latent space and reconstructs the high-dimensional citywide traffic flow from the given elements. With the help of the disentangled representations, the decomposed feature vector in latent space discloses the relationship between underlying patterns and citywide traffic dynamics. Liu Y et al. [10] propose a novel travel mode recommendation system for multi-modal transportation. In the proposed model, the feature engineering focuses on the application scenario of the multi-modal transportation recommendation and is designed from multiple perspectives of users, travel modes, locations, and time.

## B. Neural network

Neural networks' non-linear modeling ability has made them more popular. Chen CH et al. [11] proposed an arrival time prediction method (ATPM) based on recurrent neural networks (RNNs) to predict the stop-to-stop travel time for motor carriers. Pang et al. [12] proposed to exploit the long-range dependencies among the multiple time steps for bus arrival prediction via a recurrent neural network. Zhang et al. [13] proposed a model based on MapReduce combining clustering with the neural network. Yang et al. [14] proposed a novel stacked autoencoder Levenberg-Marquardt model, a type of deep architecture of neural network approach that aimed to improve forecasting accuracy. Polson et al. [15] developed a deep learning model to predict traffic flows. Wu et al. [16] proposed a deep neural network (DNN) based traffic flow prediction model (DNN-BTF) to improve the prediction accuracy. Cristina et al. [17] tried five neural network models and identified the best performing deep learning model. Wichai et al. [18] proposed another DNN-based model. Zhang et al. [19] proposed an end-to-end multitask learning temporal convolutional neural network (MTL-TCNN) to predict the short-term passenger demand at a multi-zone level. In the same year, Zhang et al. [20] also proposed a deep learning-based multitask learning (MTL) model using Bayesian optimization to tune parameters of MTL to predict short-term traffic speed. Zheng et al. [21] proposed a feature selection-based approach to identify reasonable spatial-temporal traffic patterns related to the target link

to improve online prediction performance. To resolve the problem that empirical methods cannot adequately capture various travel time distributions, Zhang et al. [22] proposed a deep learning-based Trip Information Maximizing Generative Adversarial Network (T-InfoGAN). Liu Y et al. [23] analyze the passenger flow from scopes on macroscopic and microscopic levels. Decision-tree-based models are used in modeling and predicting passenger flow. Inspired by the feature engineering of decision-tree-based models, a modular convolutional neural network is designed, which contains automatic feature extraction block, feature importance block, fully-connected block, and data fusion block. However, these articles do not combine timely GPS data with the neural network model.

## C. Support vector machine

SVM maps the input data into higher dimensional space with a specifically designed kernel such that the relationship between modified input data and the target variable is linear. Yang et al. [24] presented a prediction model of bus arrival time based on SVM with a genetic algorithm (GA-SVM). Peng et al. [25] proposed a forecasting method based on principal component analysis-genetic algorithm-support vector machine (PCA-GA-SVM) to improve arrival time prediction precision. Yao et al. [26] proposed a single-step prediction SVM model composed of spatial and temporal parameters. Moridpour et al. [27] suggested a Least Squares SVM (LS-SVM) method that expedited the training process by simplifying the quadratic programming problem using a linear regression technique. Lu LL et al. [28] propose a methodology for real-time freeway travel time estimation with data from sparse detectors, utilizing a self-organized mapping algorithm to cluster the sensors with similar traffic patterns. The data collected from the representative detectors within each cluster is then employed to estimate the travel time based on a support vector regression model.

And further, to better assist travelers in making trip decisions in under-connected environments and undersaturated signalized arterial environments, Lu LL et al. [29, 30] propose a lane-level travel time prediction model under a connected environment real-time prediction model for vehicle individuals on an undersaturated signalized arterial. Conventional models are suitable for estimating the travel time distributions of only a few road segments. In contrast, these two models fully capture travel time reliability metrics such as the buffer time index, skewness, and width of the travel time distributions for all road segments, these results will help traffic managers and engineers carry out effective traffic management and control to optimize the operation under-connected environments, and undersaturated signalized arterial environments.

## D. Particle filtering

The PF technique has been widely applied to deal with historical GPS information and predict bus arrival time. Dhivyabharathi et al. [31] proposed a method to predict stream travel time using a particle filtering approach that considers the predicted stream travel time as the sum of the median of historical travel times, random variations in travel time over time, and a model evolution error. Dhivyabharathi et al. [32] developed a model based on particle filtering technique which was better than the existing method with MAPE values around 17% with the accuracy of +/- 2 minutes, wherein inputs were obtained using the K-NN (k-nearest neighbors) algorithm (the core of KNN is that a sample belongs to most categories of k samples adjacent to it). However, the particle filtering algorithm used in these two papers is only suitable for a non-linear stochastic system with the state-space model, but the time property of bus arrival prediction was not considered.

### E. Kalman filtering

Kalman first proposed Kalman filtering in 1960. It takes the minimum mean square error as the best estimation criterion to seek a set of recursive estimation algorithms. Its basic idea is to use the state-space model of signal and noise, update the estimation of state variables by using the estimated value of the previous time and the observed value of the current time, and obtain the estimated value of the present time. It is suitable for real-time processing and computer operation. Kalman filters are ideal for conventional urban transit systems, a multi-parameter and time-varying complex giant system with a high degree of uncertainty and are non-linear. This filter can consider quantities that are partially or entirely neglected in other methods (such as the variance of the initial estimate of the state and the variance of the model error). It provides information about the quality of the estimation by providing, in addition to the best estimate, the variance of the estimation error.

KF has been widely applied to the prediction of travel time. Li et al. [33] considered KF combined with other methods and proposed a three-stage mixed model including K-means, real-time adjusted Kalman filter, and Markov historical transfer model. Huo JB et al. [34] propose a spatiotemporal extended Kalman filter (SEKF)short-term pedestrian density estimation and prediction method based on mobile phone data. A massive mobile phone dataset collected in Nanjing, China, is used in the case study. The estimated pedestrian density from Monday to Thursday is used for Friday's prediction. KF model has the advantages of high prediction accuracy and a large amount of data processed by computer. At the same time, there still is room for improvement in model construction and data input in previous studies.

While the models described in this section can solve the bus-to-station prediction problem to some degree, the influence factors these models considered are one-sided. GPS over-emphasizes the current state of the bus, degrading the prediction accuracy as the predicted distance increases. The input used in the NN network is too one-sided and does not consider the comprehensive impact of time and space characteristics. SVM relies too much on kernel tricks to achieve predictions on a large scale. The PF only finds the time of the bus to the stop and ignores the spatial effect of the bus.

The system contains accurate mathematical noise statistics (noise mean and variance matrix), which is the essential requirement of the Kalman filtering method. However, in practical application, the mathematical model and/or noise statistics of the system are always unknown (the model or noise statistics of the system contain errors). In addition, there is often model uncertainty and/or interference uncertainty; that is, there is an unmodeled dynamic system, which will cause the filtering performance to deteriorate or even diverge. This paper tries to achieve as follows:

1. Build the model to match the actual situation;

2. Noise convergence is ensured based on multi-source data input;

The above two points ensure that the performance of the Kalman filter is good to overcome the shortcoming mentioned.

This paper applies models to the public transit network data of Madison, Wisconsin, USA. It combines the Kalman filter model with the AVL and IC card intelligent bus technologies, both currently widely used, to predict urban public transit travel time.

## III. Methodology

### A. Definitions

**1) Research object.**   In conventional urban transit, the uncertainty of many external factors makes it challenging to execute vehicle and personnel scheduling and dispatching, leading

to the low reliability of public transportation. Bus ridership is consistently at low levels due to lack of information about wait times, anxiety regarding the timeliness of the bus service, and failure to arrive at the destination at the expected time. These deficiencies inconvenience urban commuters who rely on mass public transit and hinder the resultant urban traffic structures that would lower energy consumption and pollution, raise efficiency, and yield more sustainable development. Given existing technical conditions, in-depth research on the prediction method of travel time of conventional transit is of great practical value.

**2) Short-term travel time predictions.** Short-term travel time predictions are influenced by random factors (road conditions, traffic conditions, time, climate, emergencies, vehicle conditions, drivers, passengers, intersection, and control mode [35–37]). Unexpected and accidental events occur less predictably, thus requiring greater precision than long-term prediction. In this paper, the short time travel prediction is the next step prediction of travel time, that is, at time T, a short-term prediction is made for the travel time of the following decision moment $T + \Delta t$, where the prediction period $\Delta t$ is generally not more than 35 minutes.

Historical data is used to predict long-term travel time (35 minutes or more). However, travel time prediction (15 minutes or less) uses real-time data. Short-term travel time prediction (15 minutes ~ 35 minutes) uses historical and real-time data, where the recorded data is used to build and test models, and real-time information is used for online prediction and evaluation. The classification of travel time prediction is shown in Fig 1.

From this analysis, this paper's specific research focus is the short-term travel time prediction of the urban conventional transit with fixed routes and stops within the time range of 15min ~ 35min, where the system is equipped with at least advanced intelligent public transit technologies, including AVL, IC, and another real-time spatial positioning, monitoring, and transmission.

**3) Definition of bus travel time.** In this paper, public transit travel time consists of two parts: Route travel Time (RT) + stop Dwell Time (DWT). Transfer time is not included in the scope of this paper. Combined with the above discussion, this paper's definition of the public transport travel time can be expanded to the short-time travel time (15min ~ 35min) with the regular bus as the carrier, namely the single bus travel time prediction based on single-line detection.

## B. Data preprocessing

**1) The data type.** Relative to IC and dispatching information, route and stop information is relatively fixed and does not change often. It is usually obtained directly from bus companies or government passenger transport authorities. However, this information is often neither detailed nor accurate enough, especially the location of stops and other information. Discrepancies often arise in routes, requiring a supplementary survey of the routes and stops to determine their correct directions and locations.

The Madison bureau of the Wisconsin Traffic Operations and Safety Laboratory provided the data analyzed herein (see S3 File, Figs 1 and 2). All data obtained for routes and stops are spatial data. This data was digitized and stored in a spatial database utilizing GIS technology for storage. The raw data required for data mining are:

1. **Static primary data of transit**

- Route data: route number, number of stops, stop location (latitude and longitude), etc.

- Stop data: stop ID, stop site (latitude and longitude), the distance between adjacent stops, etc.

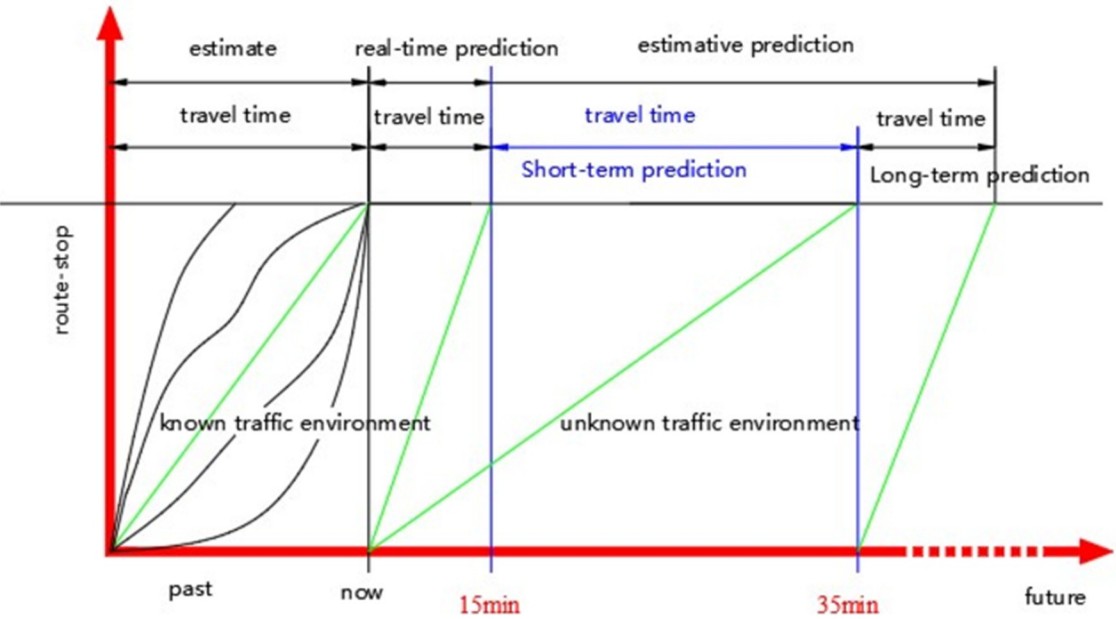

**Fig 1. Differentiation of travel time.**

2. **Dynamic primary data of transit**

- IC data: Service day, bus code, swiping time, latitude and longitude when swiping, ID of IC, route number, etc.

- AVL data: Service day, bus code, pattern ID, actual arrival time, actual departure time, time point ID, stop ID, etc.

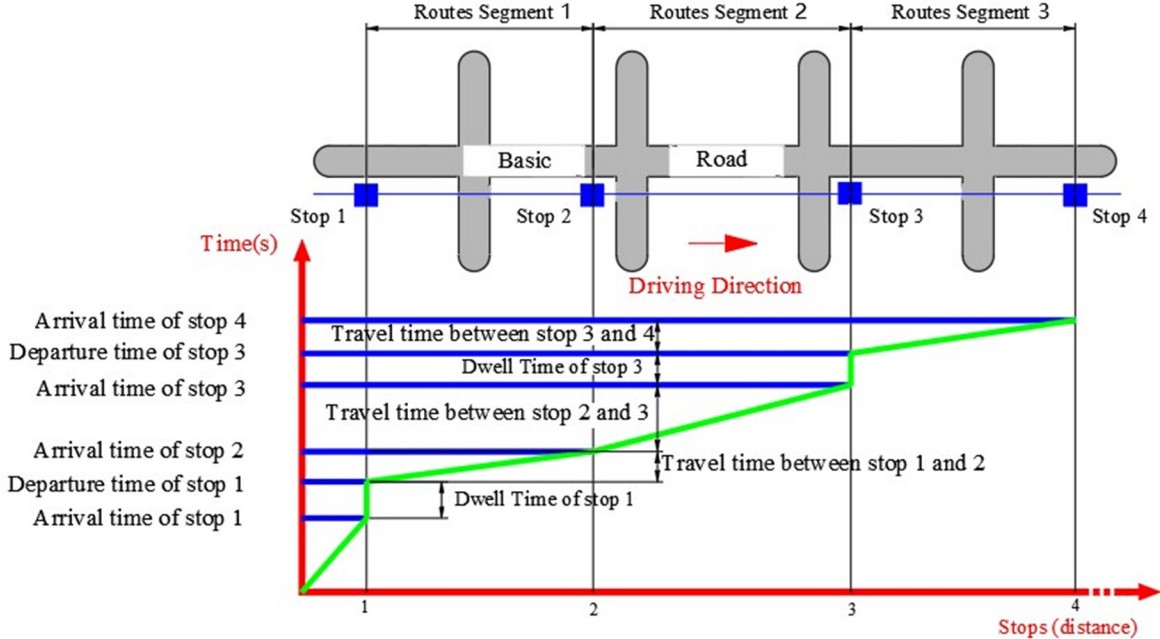

**Fig 2. Composition of bus travel time.**

**Table 1. Average absolute error and relative error of travel time prediction of the route segment.**

| Bus number | Morning peak(s) | | | Evening peak(s) | | | Flat peak(s) | | | |
|---|---|---|---|---|---|---|---|---|---|---|
| | 1 | 2 | 3 | 1 | 2 | 3 | 1 | 2 | 3 | 4 |
| Absolute error (seconds) | 11.78 | **11.20** | 11.60 | 14.11 | 13.54 | **15.98** | 11.67 | 13.12 | 12.32 | 11.56 |
| Mean relative error (seconds) | 12.0% | **9.0%** | 10.0% | 14.0% | 13.0% | **15.0%** | 12.0% | 13.0% | 12.0% | 9.0% |

■ Bus dispatching data: bus code, pattern ID, departure time, arrival time, departure interval, etc.

**2) Data preprocessing.** In this paper, data preprocessing and analysis were carried out using Oracle and MATLAB software. Clustering was carried out on the IC data, and the travel time after clustering was divided into sections to obtain the arrival, departure, and stop dwell times of each stop. These steps are shown in S1 File. The dwell time of each stop was assessed to estimate each boarding stop. Finally, the essential travel time of stops after segmentation based on the actual stops was obtained as the input data of the travel time prediction model. These results are shown in S2 File, Table 1.

AVL data was used to verify the validity of the IC data processing procedures. These steps are shown in S1 File. AVL data was also used both to estimate the parameters of the prediction model and evaluate the accuracy of the prediction model. AVL key stops (Timepoint) information was used to assess the accuracy of the segmented results: AVL key stop (Timepoint) data was used to adjust the parameters of the data processing procedures, allowing the absolute error to be minimized.

## C. Prediction model

### 1) Model assumptions.

### i. The basic assumptions

A Transit Network consists of routes and stops. Routes are comprised of route segments and stops.

1. The whole route is simplified into two parts: stop and route segment (see Fig 2). The route segment is divided into two adjacent stops: the initial and terminal stops. The route segment may or may not contain intersections. The total travel time consists of two parts: route travel time (RT) + stop dwell time (DWT).

2. All vehicles are equipped with IC and AVL systems.

3. Intersection delays and road segment delays are reflected in RT. The time of arrival and departure stop, as well as stop delays, are reflected in DWT.

4. Due to the limitation of route and dispatch, drivers are not affected by their subjective will factors.

5. The vehicle type and performance are the same on the same route.

6. There will be no changes to road infrastructure or stops during the prediction period.

7. Assuming that the bus has two doors, one door for boarding and another for disembarking, the time taken by unit passengers to get on and off is the same, and the number of people determines the time to get on and off.

### ii. Basic principle

The travel time prediction model based on single route detection comprises the route travel time prediction model (RTM), and the stop dwell time prediction model (DTM).

Route travel time is predicted by the route travel time prediction model (RTM). Each route segment's historical travel time data is used as the input to the prediction model. The arrival time at the stop for each route segment is obtained from IC card data, and the updated travel time is used to predict the new travel time, then stored as historical data. The dwell time model (DTM) and passenger arrival rate prediction model (PARM) are based on IC records of getting on and off the bus at each stop (combined with historical and current data).

The chosen historical data input for this model is the travel time data of the same period for the previous three days and the previous period of the same day, and the model output is the travel time of the current period of the day.

**2) Overall model.**   The single route travel time between stops is composed of two parts: road travel time **(RT)** + stop dwell time **(DWT)**, which can be expressed by the arrival time of buses:

$$AT_{n(i+1)} = DT_{n(i)} + RT_{n(i,i+1)} \tag{Eq 1}$$

Where,

$AT_{n(i+1)}$ is the predicted time the bus $n$ to arrives at the stop $i$ +1;

$DT_{n(i)}$ is the time the bus $n$ to leave the stop $i$;

$RT_{n(i, i+1)}$ is the travel time of bus $n$ between stops $i$ and $i$ +1 predicted by the Kalman filtering algorithm.

The stop dwell time can be expressed as:

$$DWT_{n(i+1)} = \lambda_{(i+1)} \times H_{(i+1)} \times \rho_{avg(i+1)} \tag{Eq 2}$$

Where,

$DWT_{n(i+1)}$ is the time of bus $n$ stay at stop $i$+1 to be predicted;

$\lambda_{(i+1)}$ is the passenger arrival rate at the stop $i$ +1 predicted by the Kalman filtering algorithm;

$H_{(i+1)}$ is the headway of bus $n$ in the stop $i$ + 1, $H_{(i+1)} = AT_{n(i+1)} - AT_{n-1(i+1)}$.

$\rho_{avg(i+1)}$ is consumed at the boarding time of the average passenger at the stop $i$ +1, assumed to be 2.5 seconds per person.

The time when the bus leaves the stop can be expressed as:

$$DT_{n(i+1)} = AT_{n(i+1)} + DWT_{n(i+1)} \tag{Eq 3}$$

Where,

$DT_{n(i+1)}$ is the predicted time for bus $n$ to leave the stop $i$ +1;

$AT_{n(i+1)}$ is the time of bus $n$ arriving at stop $i$ +1 predicted by Eq (1);

$DWT_{n(i+1)}$ is the time of bus $n$ stay at stop $i$ +1 predicted by Eq (2).

Meanwhile, according to Eqs (1), (2) and (3), the time of departure from the stop $i$ +1 is:

$$DT_{n(i+1)} = DT_{n(i)} + RT_{n(i,i+1)} + \lambda_{(i+1)} \times H_{(i+1)} \times \rho_{avg(i+1)} \tag{Eq 4}$$

In the equation, each parameter is the same as the above equation. The above equations can predict the arrival and departure times of bus $n$ at stop $i$+1. In the equation, $RT_{n(i, i+1)}$ and $\lambda_{i+1}$ are obtained from the Route Travel Time Prediction Model(RTM) and the Passenger Arrival Rate Prediction Model(PARM).

### i. Route Travel Time Prediction Model (RTM)

The road travel time prediction model of period $k$+1 of a single route between adjacent stops can be expressed in Eqs (5) ~ (8):

$$g(k+1) = \frac{e(k) + VAR(data_{out})}{e(k) + VAR(data_{out}) + VAR(data_{in})} \tag{Eq 5}$$

$$a(k+1) = 1 - g(k+1) \qquad \text{(Eq 6)}$$

$$e(k+1) = VAR(data_{in}) \times g(k+1) \qquad \text{(Eq 7)}$$

$$RT_{n(i,i+1)}(k+1) = a(k+1) \times art(k) + g(k+1) \times Avg(art)$$

$$= a(k+1) \times art(k) + g(k+1) \times \frac{art_1(k+1) + art_2(k+1) + art_3(k+1)}{3} \qquad \text{(Eq 8)}$$

Where,

$g\,(k+1)$ is the gain of the Kalman filter in the period $k+1$;

$a\,(k+1)$ is the loop gain in the period k +1;

$e\,(k)$ is the filter error in period $k$, which is calculated in the previous cycle;

$e\,(k+1)$ is the filter error of period $k+1$, which is used for the calculation of the following process;

$RT_{n(i,i+1)}(k+1)$ is the travel time of bus $n$ between stops $i$ and $i+1$ predicted by Kalman filtering algorithm in period $k+1$;

$art\,(k)$ is the travel time of bus $n$ between stop $i$ and $i+1$ in period $k$;

$art_1\,(k+1)$ is the travel time of bus $n$ between stops $i$ and $i+1$ during period $k+1$ on the previous day;

$art_2\,(k+1)$ is the travel time of bus $n$ between stops $i$ and $i+1$ during period k+1 on the previous two days;

$art_3\,(k+1)$ is the travel time of bus $n$ between stops $i$ and $i+1$ during period k+1 on the previous three days;

$VAR(data_{out})$ is the variance of the prediction;

$VAR(data_{in})$ is the variance of the travel time "$art_1\,(k+1)$, $art_2\,(k+1)$, $art_3\,(k+1)$" between stops $i$ and $i+1$ for the previous three days, which can be expressed by the travel time " $art_1\,(k+1)$, $art_2\,(k+1)$, $art_3\,(k+1)$"of the previous three days:

$$VAR\,(data_{in}) = VAR[art_1(k+1), art_2(k+1), art_3(k+1)] \qquad \text{(Eq 9)}$$

The variance of the random variable can be expressed as:

$$VAR(X) = E[X - E[X])^2] \qquad \text{(Eq 10)}$$

$$E\,(X) = Avg(art) = \frac{art_1(k+1) + art_2(k+1) + art_3(k+1)}{3} \qquad \text{(Eq 11)}$$

The equation expresses the relevant variables:

$$\Delta_1 = [art_1(k+1) - avg(art)]^2 \qquad \text{(Eq 12)}$$

$$\Delta_2 = [art_2(k+1) - avg(art)]^2 \qquad \text{(Eq 13)}$$

$$\Delta_3 = [art_3(k+1) - avg(art)]^2 \qquad \text{(Eq 14)}$$

$$VAR(data_{in}) = \frac{\Delta_1 + \Delta_2 + \Delta_3}{3} \qquad \text{(Eq 15)}$$

**Table 2. Absolute and relative errors of the total travel time prediction of the whole route.**

| Bus number | Morning peak(s) | | | Evening peak(s) | | | Flat peak(s) | | | |
|---|---|---|---|---|---|---|---|---|---|---|
| | **1** | **2** | **3** | **1** | **2** | **3** | **1** | **2** | **3** | **4** |
| Absolute error (minutes) | 1.23 | 1.98 | 2.11 | 0.98 | **2.23** | 0.78 | 1.43 | **0.17** | 0.83 | 1.76 |
| The relative error | 2.3% | **0.4%** | 3.0% | 4.0% | 6.0% | 4.0% | 3.0% | 2.2% | **7.0%** | 4.0% |

$VAR(data_{out})$ is determined by the prediction result of the filter model and the observation value in the future, which cannot be obtained because the prediction result is unknown, and the future trip has not yet happened, ideally, under the condition of good prediction state, $VAR(data_{out}) = VAR(data_{in})$, the introduction of new variables: $VAR(local_{data})$ is used to indicate the $VAR(data_{out})$, $VAR(data_{in})$ and can be represented as:

$$VAR(local_{data}) = VAR(data_{in}) = VAR(data_{out}) \qquad \text{(Eq 16)}$$

Filter gain (Eq (5) and filter error (Eq (7)) can be reduced to Eqs (17) and (18):

$$g(k+1) = \frac{e(k) + VAR(local_{data})}{e(k) + 2VAR(local_{data})} \qquad \text{(Eq 17)}$$

$$e(k+1) = VAR(local_{data}) \times g(k+1) \qquad \text{(Eq 18)}$$

From what has been discussed above, the prediction model of route travel time for a single route between stops based on the Kalman filter mainly consists of Eqs (6), (8), (17) and (18). The four essential equations can predict the route travel time of bus $n$ on the whole route by rolling(these results are shown in S2 File, Table 2).

### ii. Stop Dwell Time prediction Model (DTM)

#### (1) Stop Dwell Time Prediction Model

Stop dwell time refers to the bus's time at a stop due to events. According to the vehicle arrival process, stop dwell time includes passenger service time, acceleration and deceleration time, opening and closing door time, and additional delay time caused by queuing at the stop. Most passenger service time is when passengers require to board and disembark the bus. Acceleration and deceleration times, opening and closing door times are the same for the same type of bus on the same route. Additional delays caused by queuing occur at stops with multiple routes, and there are second dwell at longer stops. However, direct observation finds that when there is a queue at the stop, buses can often use the queuing time to complete the loading and unloading service. The time lost in queuing is considered negligible due to its small size.

Therefore, this paper only considers the difference in service time of passengers on and off the bus. The dwell time of the bus at the stop $i+1$ is mainly determined by the number of passengers arriving in a unit time (expressed by Eq (2)). The equation shows that $H(i+1)$ can be obtained directly from the IC data. The only predictive value in this equation is the passenger arrival rate $\lambda_{(i+1)}$.

#### (2) Passenger Arrival Rate prediction Model (PARM)

$\lambda_{(i+1)}$ is the number of passengers arriving per unit of time. The historical arrival rate $\lambda_{(i+1)}$ per stop can be obtained through IC data mining technology. In this paper, the Kalman filtering method is still used to predict the passenger arrival rate of stop $i+1$. The prediction model

construction follows, $VAR(local_{data})$ can be expressed as follows:

$$VAR(local_{data}) = VAR[par_1(k+1), \ par_2(k+1), \ par_3(k+1)] \qquad \text{(Eq 19)}$$

There are:

$$E(X) = Avg(par) = \frac{par_1(k+1) + par_2(k+1) + par_3(k+1)}{3} \qquad \text{(Eq 20)}$$

$$\Delta_1 = [par_1(k+1) - avg(par)]^2 \qquad \text{(Eq 21)}$$

$$\Delta_1 = [par_2(k+1) - avg(par)]^2 \qquad \text{(Eq 22)}$$

$$\Delta_1 = [par_3(k+1) - avg(par)]^2 \qquad \text{(Eq 23)}$$

$$VAR(local_{data}) = \frac{\Delta_1 + \Delta_2 + \Delta_3}{3} \qquad \text{(Eq 24)}$$

The Kalman gain $g \ (k+1)$ and the loop gain $a \ (k+1)$, $\lambda_{(i+1)}(k+1)$ prediction model:

$$g(k+1) = \frac{e(k) + VAR(local_{data})}{e(k) + 2VAR(local_{data})} \qquad \text{(Eq 25)}$$

$$a(k+1) = 1 - g(k+1) \qquad \text{(Eq 26)}$$

$$\lambda_{(i+1)}(k+1) = a(k+1) \times par(k) + g(k+1) \times Avg(par)$$
$$= a(k+1) \times par(k) + g(k+1) \times \frac{par_1(k+1) + par_2(k+1) + par_3(k+1)}{3} \qquad \text{(Eq 27)}$$

Where,

$g \ (k+1)$ is the gain of the Kalman filter in the period $k$ +1;

$a \ (k+1)$ is the loop gain in the period $k$ +1;

$e \ (k)$ is the filter error of period $k$, which is calculated from the previous cycle;

$e \ (k+1)$ is the filter error of period $k$ +1, which is used for the calculation of the following process;

$par \ (k)$ is the passenger arrival rate between stops $i$ and $i$ +1 during the period $k$;

$par_1 \ (k+1)$ is the passenger arrival rate at stop $i$+1 during period $k$ +1 in the previous day;

$par_2 \ (k+1)$ is the passenger arrival rate at stop $i$+1 during period $k$ +1 in the previous two days;

$par_3 \ (k+1)$ is the passenger arrival rate at stop $i$+1 during period $k$ +1 in the previous three days.

$VAR(local_{data})$ is the variance of *the* passenger arrival rate of "$par_1 \ (k+1)$, $par_2 \ (k+1)$, $par_3 \ (k$ +1)" during period $k$+1 between stops $i$ and $i$ +1 in the previous three days.

$$e(k+1) = VAR(local_{data}) \times g(k+1) \qquad \text{(Eq 28)}$$

The Passenger Arrival Rate prediction Model (PARM) based on the Kalman filter is mainly composed of Eqs (25), (26), (27), and (28). Based on the above four essential equations, the passenger arrival rate of bus $n$ in the period $i$ +1 can be predicted by rolling to obtain the dwell time of each stop (these results are shown in S2 File, Table 3).

**3) Prediction process.** In conclusion, the travel time prediction model constructed above can predict the travel time of the whole route.

Table 3. The mean absolute error of predictive travel time of route 2 # (entire route).

| Route | Mean Absolute Error (MAE, S) | |
|---|---|---|
| | Prediction of arrival time | Prediction of departure time |
| #2 | 59.24 | 46.41 |

**Step 1**: **Overall prediction**

1. Determine the target route predicted and its initial and terminal stops: determine the initial forecast period $k$ +1, assume stop $i$ as the initial stop, and predict the time of bus $n$ arriving at stop $i$ +$n$;

2. Prepare the basic data required for prediction;

3. $i = 0$;

4. $i = i+1$;

5. Executing Stop Dwell time prediction Module (DTM): to predict the dwell time of bus $n$ at stop $i$, $DWT_{n(i)}$;

6. Executing Route Travel time prediction Module (RTM): to predict the route travel time of bus $n$ between stop $i$ and $i+1$ during the $k+1$ period, $RT_{n(i,\ i+1)}(k+1)$;

7. Sum, $DT_{n(i)} = AT_{n(i)}+DWT_{n(i)}$, $AT_{n(i+1)} = DT_{n(i)} = RT_{n(i,\ i+1)}$;

8. Determine if $i$ +1 = $n$, then end the operation; otherwise, return to Step (4) and continue the procedure until the end of the prediction.

After the repeated cyclic operation, the arrival time of bus $n$ $AT_{n(i+n)}$ at stop $i$ +$n$ during period $k+m$ can be obtained by stop, one by one (as shown in Fig 3).

**Step 2–1**: **Built-in module (Stop Dwell Time Prediction Module (DTM))**

1. Initialize $e\ (k) = 0$, assuming stop $i$ is the initial stop;

2. The historical data of passenger arrival rate between stop $i$ and $i$ +1 for these four days (during period $k$ +1 in the previous three days, during period $k$ in the same day) can be obtained through the IC card system: $par_3\ (k+1)$, $par_2\ (k+1)$, $par_1\ (k+1)$, $par(k)$;

3. The passenger arrival rate of stop $i$ +1 in period $k$ +1 can be predicted according to the Eqs (25), (26), (27), and (28);

4. The headway of the corresponding stop in period $k+1$ of the predicted day can be obtained through the bus IC data;

5. According to Eq (2), the dwell time of stop $i$ +1 in period $k+1$ can be predicted.

**Step 2–2**: **Built-in module (Route Travel Time Prediction Module (DTM))**

1. Initialize $e\ (k) = 0$, assuming stop $i$ is the initial stop, and the AVL system can obtain its departure time, and it is set as $DT_{n(i)}$;

2. The historical data of travel time between stop $i$ and $i$ +1 for these four days (during period $k$ +1 in the previous three days, during period $k$ in the same day) can be obtained through the IC card system: $art_3\ (k+1)$, $art_2\ (k+1)$, $art_1\ (k+1)$, $art(k)$;

3. According to Eqs (6), (8), (17), and (18), the route travel time of bus $n$ between stop $i$ and $i$ +1 in period $k+1$ can be predicted $RT_{n(i,\ i+1)}(k+1)$.

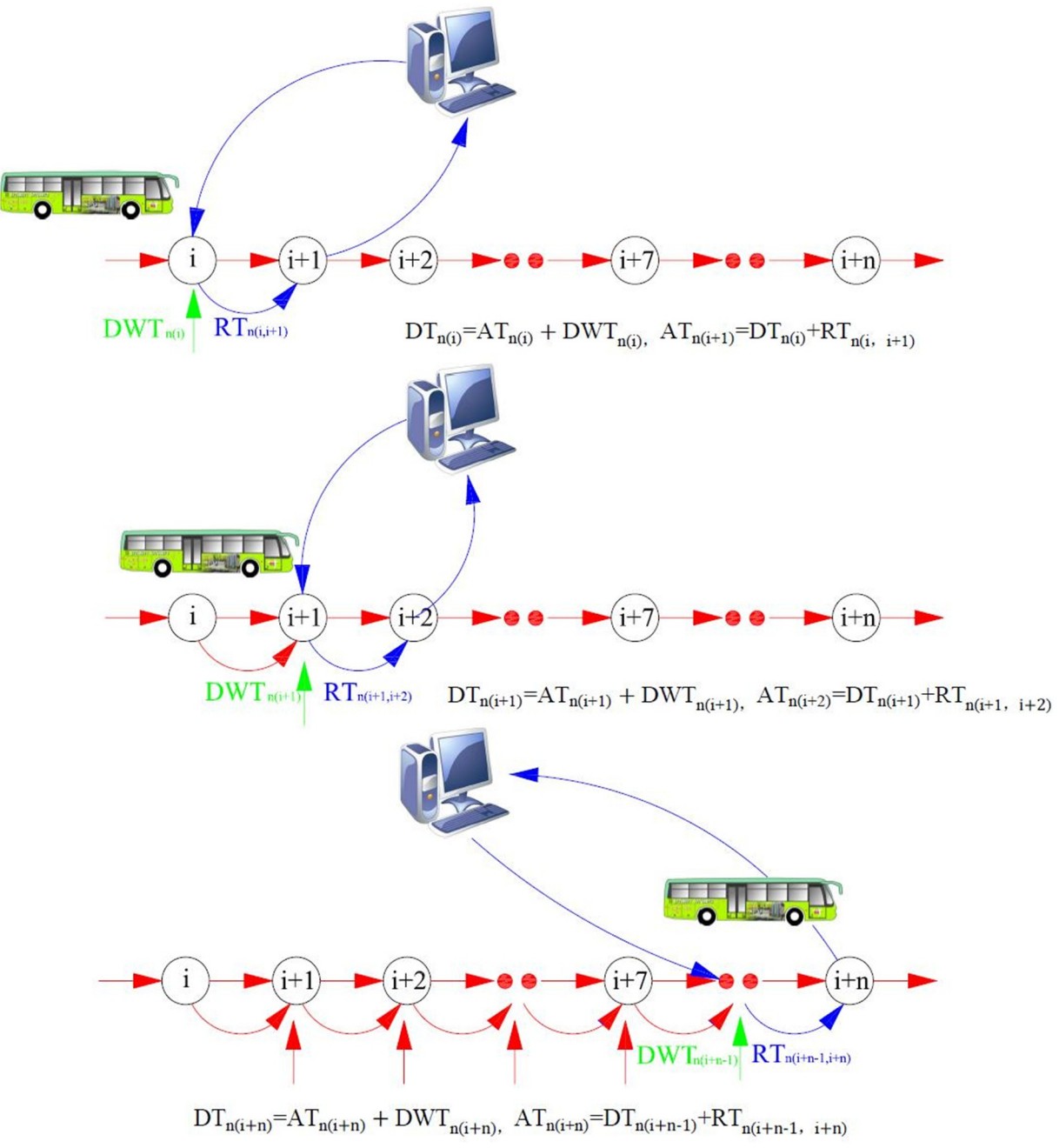

$$DT_{n(i)}=AT_{n(i)} + DWT_{n(i)}, \quad AT_{n(i+1)}=DT_{n(i)}+RT_{n(i, i+1)}$$

$$DT_{n(i+1)}=AT_{n(i+1)} + DWT_{n(i+1)}, \quad AT_{n(i+2)}=DT_{n(i+1)}+RT_{n(i+1, i+2)}$$

$$DT_{n(i+n)}=AT_{n(i+n)} + DWT_{n(i+n)}, \quad AT_{n(i+n)}=DT_{n(i+n-1)}+RT_{n(i+n-1, i+n)}$$

**Fig 3. Steps of travel time prediction between stops.**

The prediction process is shown in Fig 4.

## IV. Case study

### A. Data preprocessing

**1) Circuit information.** This example selects route 2 in Madison, Wisconsin, located in the Madison metropolitan area. The chosen route is the section between N Frances Street in

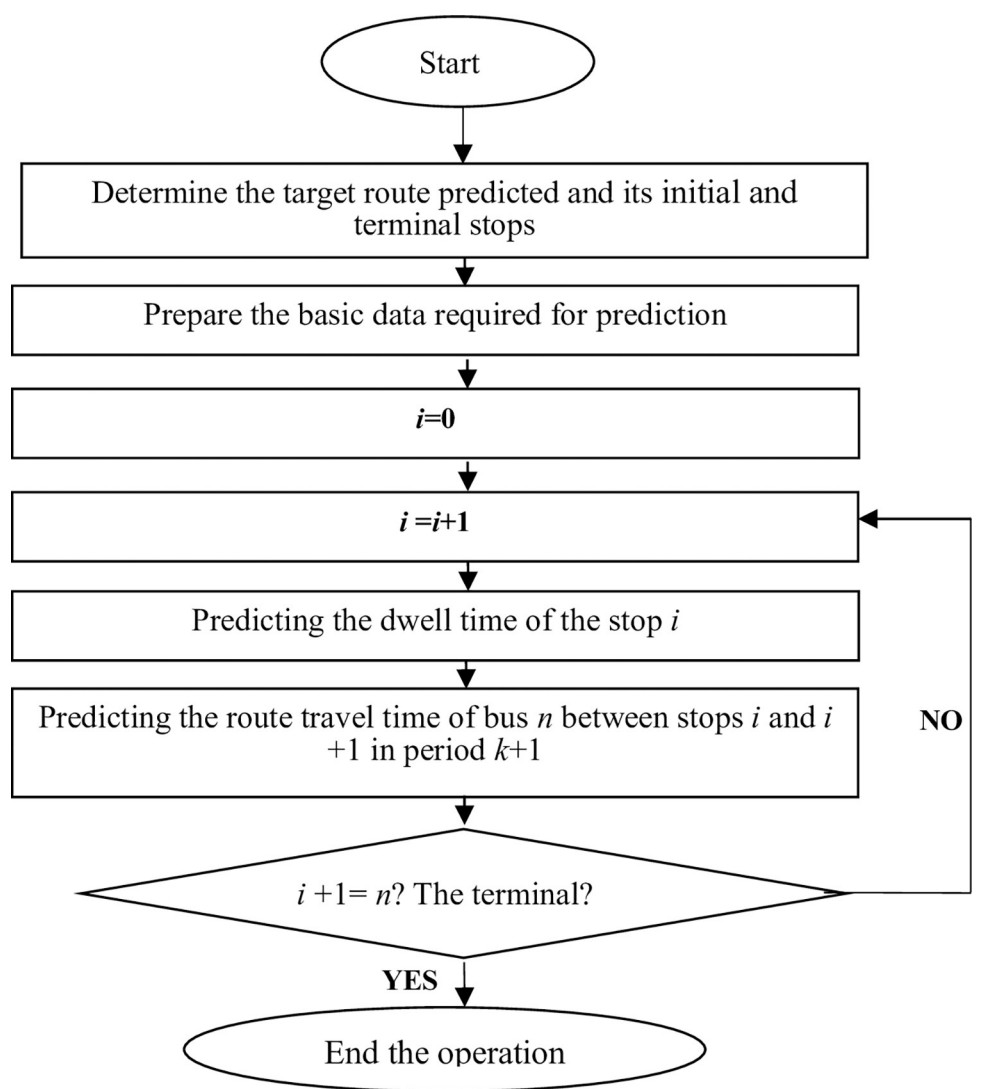

**Fig 4. Travel time prediction process based on single route detection.**

the east and N Midvale Blvd in the west and traverses the city's center. The route goes through different road types and thus can comprehensively reflect the operational characteristics of buses under other road conditions (shown in S3 File, Figs 3 and 4).

   **2) Data information.**   The IC card and AV data of four days on October 1, 2, 3, and 4, 2019 were selected as the primary data of the example, and the historical data of period $k+1$ on October 1, 2, and 3 and period $k$ on October 4 were used to predict the data of period $k+1$ on October 4. Among them, 4,157 sets of travel time data (IC data) were used as the model's input data, and 2,124 sets of AVL data were used as the validation data of the model.

## B. Preliminary results

According to the characteristics of urban roads, in this paper, the day was divided into three time periods, with morning and evening peaks as the boundary:

■ Period 1: 7:00–9:00, morning peak;

■ Period 2: 17:00–19:00, evening peak;

■ Period 3: another period, flat peak.

Given these three periods, this paper will first analyze the change laws between these three periods and then analyze the change rule within each period.

**1) Route segment between stops.** The average absolute error and relative error of the travel time prediction of the ten shifts were calculated. Table 1 shows the results.

Table 2 shows that the average absolute error of the travel time prediction between stops is concentrated between 11~16 seconds, and the average absolute error of the evening peak is slightly higher than that of the other two periods. The average relative error is higher than the whole route, concentrated in the range of 9% to 15%.

**2) Whole route.** The travel time of the whole route is the total time taken by the bus between the initial and terminal stop (prediction results are shown in S2 File, Table 4). Table 2 shows the calculations of the absolute error and relative error of the total travel time prediction of the route of 10 shifts between 128 stops in the entire day.

Table 2 shows that the absolute error of the total travel time prediction for the ten shifts is within 2.25 minutes, with the maximum absolute error of 2.23 minutes and the minimum absolute error of only 0.17 minutes. The maximum relative error is 7%, and the minimum relative error is only 0.4%. At the same time, the absolute and relative errors of the morning, evening, and flat peaks are not significantly different, indicating that the prediction model can respond to the change of traffic conditions over time and can make an accurate prediction of travel time during peak hours.

In summary, the relative error of the model is within 15%, and the absolute error is concentrated in about 11~16 seconds when the route travel time between stops is taken as the prediction object. The relative error of the model is within 7%, and the absolute error is concentrated in 0~2.5 minutes when the whole route is taken as the prediction object. The forecast method proposed in this paper can accurately predict the travel time of buses, and the prediction effect of the whole route is better than that of the route segment between stops.

**Table 4. Prediction error of travel time between stops.**

| Route Segment | The Evaluation Index | Kalman Filter | The Neural Network |
|---|---|---|---|
| Route segment 1 | MRE | 0.0681 | 0.0750 |
| | RSRE | 0.0730 | 0.0849 |
| | MARE | 0.1155 | 0.1431 |
| Route segment 2 | MRE | 0.0276 | 0.0632 |
| | RSRE | 0.0355 | 0.0740 |
| | MARE | 0.0760 | 0.1303 |
| Route segment 3 | MRE | 0.0750 | 0.1638 |
| | RSRE | 0.0918 | 0.3494 |
| | MARE | 0.2240 | 0.0661 |
| Route segment 4 | MRE | 0.0859 | 0.1076 |
| | RSRE | 0.1076 | 0.1431 |
| | MARE | 0.2290 | 0.1214 |
| Route segment 5 | MRE | 0.0434 | 0.1115 |
| | RSRE | 0.0543 | 0.2290 |
| | MARE | 0.1204 | 0.1194 |
| Route segment 6 | MRE | 0.0424 | 0.1392 |
| | RSRE | 0.0444 | 0.1303 |
| | MARE | 0.0967 | 0.2280 |

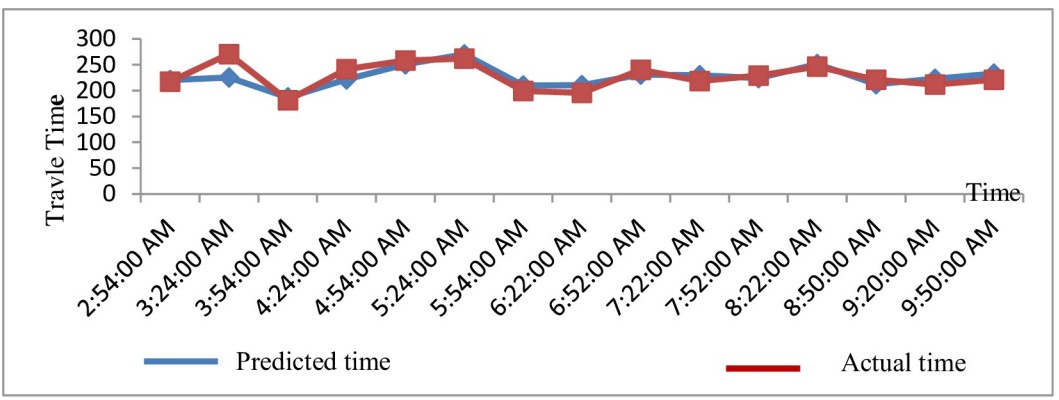

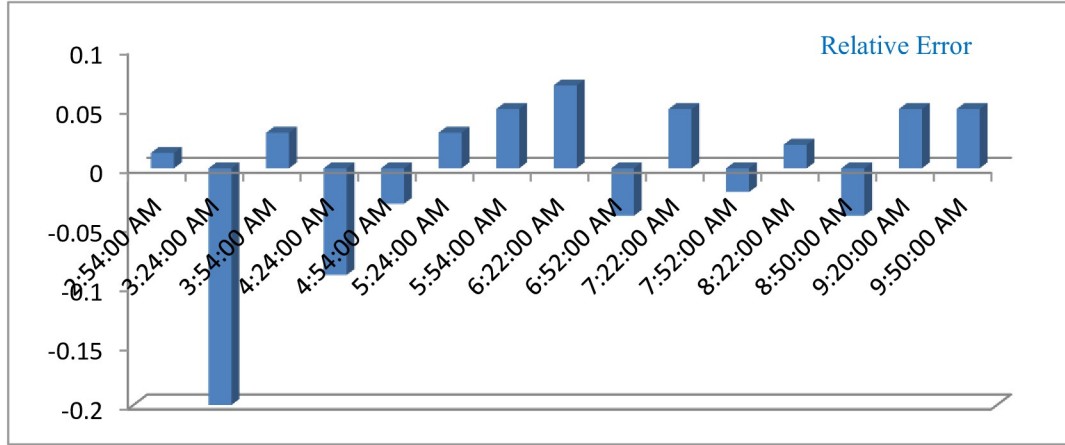

**Fig 5. Predicted result of travel time (route segment 1).**

## C. Evaluation

Travel time predictions for 235 groups of 128 sections between stops of route 2# were carried out throughout the day from 6:00 to 18:00. The following three forms were evaluated respectively in route segment between stops, whole route, and comparison with another prediction method.

**1) Route segment between stops.**

**i. Route segment 1 (CAMPUS & BABCOCK[ID#0809] W JOHNSON & CHARTER [ID#0581])**

Fig 5 shows the predicted travel time for route segment 1 using this process (Campus & Babcock [ID#0809] W Johnson & Charter [ID#0581]).

**ii. Route segment 2 (W JOHNSON & CHARTER [ID#0581] W JOHNSON & MILLS [ID#0741])**

Fig 6 shows the predicted travel time for route segment 2(W Johnson & Charter [ID #0581] W Johnson & Mills [ID #0741]).

**iii. Route segment 3 (W JOHNSON & MILLS [ID#0741] W JOHNSON & N PARK [ID#0435])**

Fig 7 shows the predicted travel time for route segment 3 (W Johnson & Mills [ID #0741] W Johnson & N Park [ID #0435]).

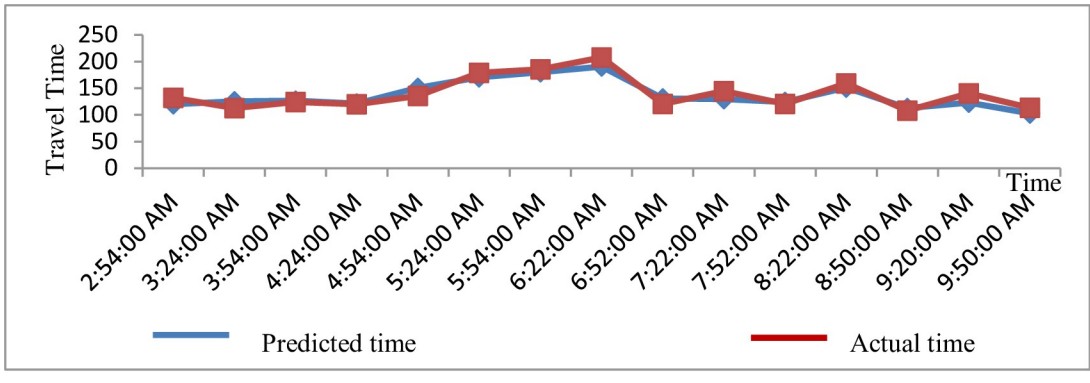

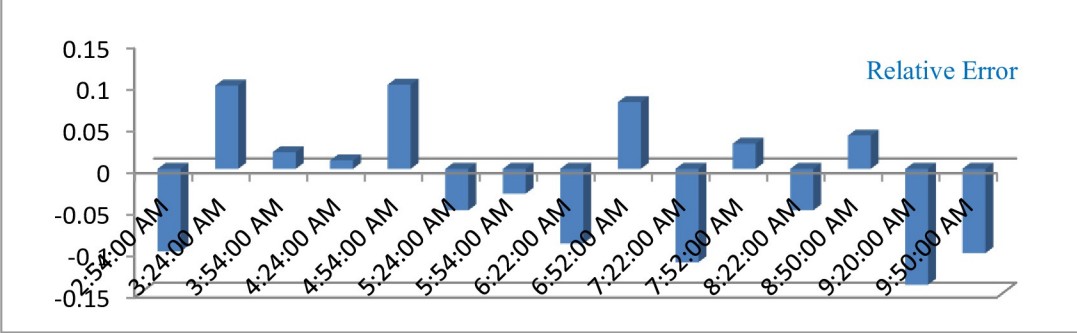

**Fig 6. Predicted result of travel time (route segment 2).**

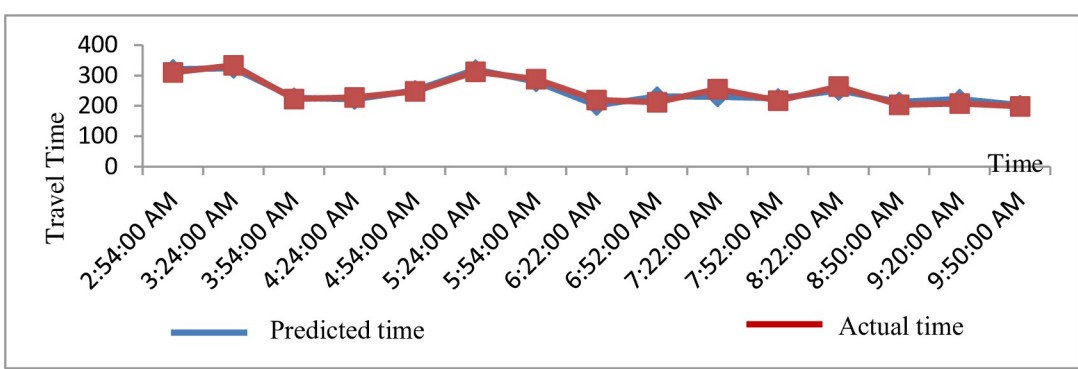

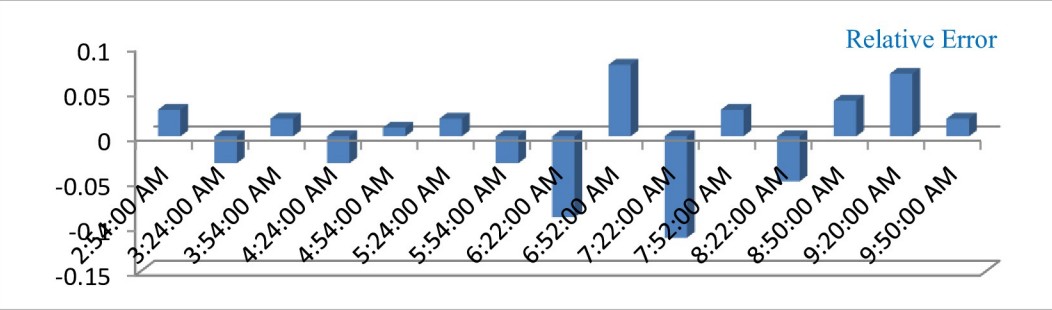

**Fig 7. Predicted result of travel time (route segment 3).**

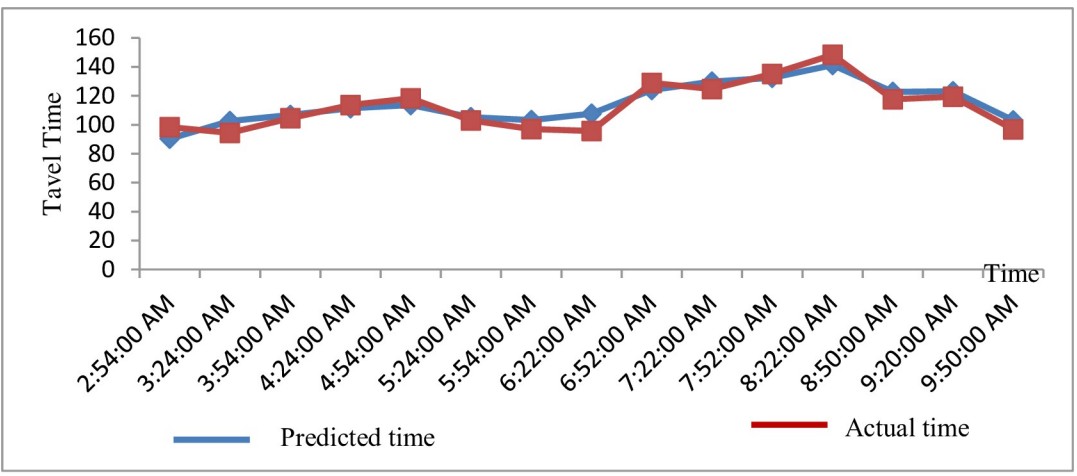

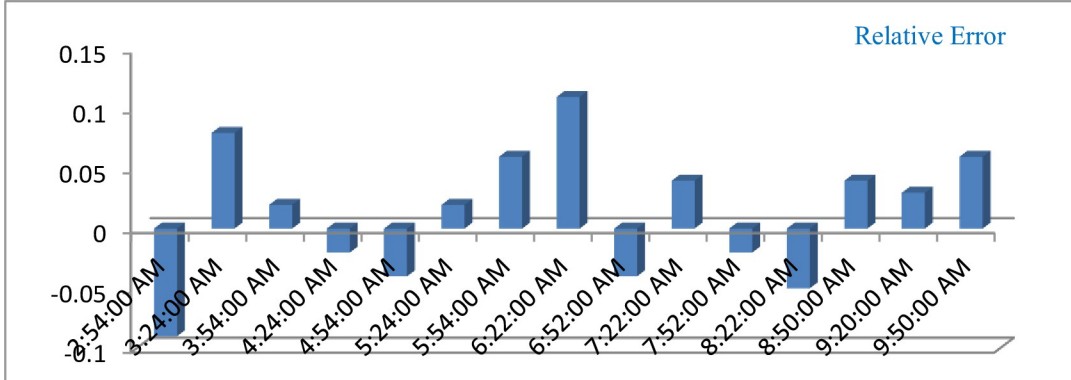

**Fig 8. Predicted result of travel time (route segment 4).**

### iv. Route segment 4 (W JOHNSON & PAKE [ID#0435] W JOHNSON & FRANCES [ID#0941])

Fig 8 shows the predicted result travel time for route segment 3 (W Johnson & Pake [ID #0435] W Johnson & Frances [ID #0941]).

**2) Whole route.** The critical stop data (time points) across route 2# was taken for analysis and evaluation. The Mean Absolute Error (MAE) of the whole route was evaluated. Table 3 displays these results:

**3) Comparison with other prediction models.** Since neural network models can guarantee real-time prediction and adequately control multiple influencing factors in public transit systems, previous researchers primarily used neural network models to predict travel time. Comparing and analyzing the current model's prediction results against neural networks, a travel time prediction method based on a neural network model was introduced. The two forecasting methods' Mean Relative Error (MRE), Root Squared Relative Error (RSRE), and Maximum Relative Error (MARE), were calculated. From the evaluation results, the MRE were all less than 20%, indicating that this model is a good prediction model. The calculation results are shown in Table 4.

Table 4 demonstrates that the Kalman filtering algorithm is more accurate and stable than the neural network, with less fluctuation and minor errors.

## IV. Conclusions

The travel time prediction of public transit is one of the effective measures to improve service reliability and travel structure, alleviate traffic problems. The Kalman filter model has high precision in one-step-ahead prediction and uses computer software to calculate massive data. Under the background of the urban intelligent public transit system and big data, this paper focuses on an urban conventional transit system. The Kalman filter-based travel time prediction technology of an urban public transit system with various random traffic factors was studied based on AVL and IC data.

1. Prediction model: The overall prediction model (including prediction steps and processes) was constructed: 1) Road travel Time prediction Model (RTM); 2) stop Dwell time prediction Model (DTM) and Passenger Arrival Rate prediction Model (PARM).

2. Model evaluation: The model's evaluation criteria and indices are given; The travel time of route #2 in Madison was selected for prediction, and the error analysis of the prediction results was carried out in combination with AVL data. The results show that the model can meet the accuracy requirements of the travel time prediction, and the prediction effect of the whole route is superior to that of the route segment between stops.

In this paper, the travel time considered only a single bus in a journey, which is the basis of passenger-oriented travel time predictions. The prediction model should be expanded to be more consistent with the actual application scenario of a conventional large-scale composite transit network. Due to the data's particularity and the demand's diversity and complexity, future research should focus on these points:

1. Data mining: In practical applications, massive multi-source data is needed. Meticulous data collection, processing, and analysis are conducive to improving the accuracy of prediction results. Therefore, the construction and management of databases need further research. Data mining is an iterative process in which mining results are constantly applied to practice, the effect is tested, and the mining algorithm is improved. Further research is needed on the actual application effect, user evaluation, and improvement direction of data mining algorithm in the intelligent public transit system to promote the perfection and improvement of the prediction model.

2. Prediction model of travel time: This paper only studies the travel time prediction method based on the Kalman filter. In future research, other commonly used prediction methods such as Long Short-Term Memory (LSTM) and comprehensive modeling can be combined to improve prediction accuracy.

3. Levels of travel time prediction: In this paper, the travel time focuses on a single bus in a journey, which is the basis of traveler-oriented travel time predictions. In contrast, a trip consists of the sum of one or more bus journeys. Based on several routes for multiple buses, travel time prediction problems involving line selection and transfer will need to consider the transit system's network capacity. For a large-scale network, the complexity will increase sharply with stops and routes. Therefore, making the method scalable to large networks should be studied.

4. Promotion and application: Extensive cooperation and investment of enterprises and research institutes would benefit the use of the models described here. Special attention to the characteristics of those local enterprises, allowing measures to be adjusted to local conditions will ease the application and promotion of this technology. Properly promoted and

applied, these models will contribute to developing and using intelligent public transit systems to meet the needs of passengers, operators, managers, and policymakers.

## Supporting information

**S1 File.**
(DOCX)

**S2 File.**
(DOCX)

**S3 File.**
(DOCX)

## Acknowledgments

We would like to acknowledge Mr. Hongjie Liu for providing the part of data for the study. Additionally, special gratitude to Mr. Les Lauber for reviewing the language usage. Thanks to anonymous reviewers for their constructive comments and suggestions regarding the earlier version of this paper.

## Author Contributions

**Conceptualization:** Xinhuan Zhang, Hongjie Liu.

**Data curation:** Xinhuan Zhang, Hongjie Liu, Meili Xie, Yuran Pan.

**Formal analysis:** Xinhuan Zhang, Hongjie Liu, Yuran Pan.

**Funding acquisition:** Xinhuan Zhang.

**Investigation:** Xinhuan Zhang, Junqing Shi.

**Methodology:** Xinhuan Zhang, Hongjie Liu.

**Project administration:** Xinhuan Zhang.

**Resources:** Xinhuan Zhang.

**Supervision:** Xinhuan Zhang, Hongjie Liu.

**Validation:** Junqing Shi.

**Writing – original draft:** Xinhuan Zhang.

**Writing – review & editing:** Les Lauber.

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
