## [Decision Letter · Decision Letter 0]

2 Aug 2021

PONE-D-21-18697

Travel Time Prediction of Urban Public Transportation Based on Detection of Single Routes

PLOS ONE

Dear Dr. Zhang,

Thank you for submitting your manuscript to PLOS ONE. After careful consideration, we feel that it has merit but does not fully meet PLOS ONE’s publication criteria as it currently stands. Therefore, we invite you to submit a revised version of the manuscript that addresses the points raised during the review process.

We look forward to receiving your revised manuscript.

Kind regards,

Jian Wang

Academic Editor

PLOS ONE

Journal Requirements:

2. Please provide further details in your methods section of the source of the dataset, how it was collected, and how it can be accessed by other researchers seeking to reproduce your study

3. We note that Figure 3 in your submission contain satellite images which may be copyrighted. All PLOS content is published under the Creative Commons Attribution License (CC BY 4.0), which means that the manuscript, images, and Supporting Information files will be freely available online, and any third party is permitted to access, download, copy, distribute, and use these materials in any way, even commercially, with proper attribution. For these reasons, we cannot publish previously copyrighted maps or satellite images created using proprietary data, such as Google software (Google Maps, Street View, and Earth). For more information, see our copyright guidelines: http://journals.plos.org/plosone/s/licenses-and-copyright.

1. You may seek permission from the original copyright holder of Figure 3 to publish the content specifically under the CC BY 4.0 license.  

4. We suggest you thoroughly copyedit your manuscript for language usage, spelling, and grammar. If you do not know anyone who can help you do this, you may wish to consider employing a professional scientific editing service. 

Additional Editor Comments (if provided):

Please note that one reviewer reject this paper and two reviewers suggest minor revision. I would like to make a decision of major revision. Please carefully revise the manuscript and provide the responses to the reviewers. Further, the manuscript contains many grammar mistakes and typos. A professional English editing is suggested.

Reviewers' comments:

Reviewer's Responses to Questions

**Comments to the Author**

1. Is the manuscript technically sound, and do the data support the conclusions?

Reviewer #1: Partly

Reviewer #2: Yes

Reviewer #3: Yes

2. Has the statistical analysis been performed appropriately and rigorously? 

Reviewer #1: N/A

Reviewer #2: Yes

Reviewer #3: Yes

3. Have the authors made all data underlying the findings in their manuscript fully available?

Reviewer #1: No

Reviewer #2: Yes

Reviewer #3: Yes

4. Is the manuscript presented in an intelligible fashion and written in standard English?

Reviewer #1: Yes

Reviewer #2: Yes

Reviewer #3: Yes

5. Review Comments to the Author

Reviewer #1: This article develops a Kalman filter-based travel time prediction of urban public transportation model.

The reviewer has the following comments:

1) Please remove all comments and be professional!

2) In Section II, there is a syntax error in ‘This literature review presents an overview of methods for predicting travel time published in the last 5 years is presented’.

3) The flow in Section II would confuse readers a lot. Why not move ‘Section II.B Kalman filtering’ after the other four sub-sections, as this paper chooses Kalman filtering as the optimal algorithm? Present the pros and cons of SVM, GPS, PF, and NN, and then articulate the advantage of Kalman filter over others.

4) The research gaps are still unclear. Sections I & II states the reason why this paper choose Kalman filter as the tool for travel time prediction, but none of current research gaps has been presented.

5) The potential contributions and renovations of this paper are still unclear by the end of Section II. Please add potential contributions and renovations at the end of Section II.

6) The statement at the end of Section III.A.1), ‘Given existing technical conditions, in-depth research on the prediction method of travel time of conventional transit (especially conventional transit under mixed traffic conditions) is of great theoretical significance and application value’, is inconvincible, as this paper never mentions anything related to mixed traffic before or existing technical conditions.

7) What is ‘random factor’ in Section III. A. 2). Please add definitions to terms before you use them.

8) The flow of this paper is very very hard to follow. There is no connection between sub-sections and even paragraphs.

9) The writing of this paper is very unfriendly to readers.

10) Recommend authors to well re-organize the structure of this paper, improve the academic writing, and rephrase the whole paper before submission.

Reviewer #2: This paper combines the Kalman filter model with the AVL and IC card intelligent bus technologies to predict the travel time of urban public transport. It is an interesting work. However there are some key problems need to be addressed:

1. The part of Literature review lacks research contributions and the motivation of choosing Kalman filter model.

2. The detailed parameters and processes of neural network travel time prediction model should be explained, since current comparisons are insufficient.

3. Some data (such as bus code, pattern ID, departure time, et al.) has been detected with multiple devices (i.e., IC, AVL, and Bus dispatching). The authors should demonstrate the fusion process of these data.

4. The parts of Abstract and Conclusions should add the methodology and results with real engineering of the research.

5. Some minor writing mistake “Kilman” should be “Kalman”, “△” should be “Δ”, not triangle, et al.

Reviewer #3: Based on Automatic Vehicle Location (AVL) and IC data, this paper proposed a variety of random traffic factors, constructed transit travel time prediction model and carried out comparative analysis on the actual results. The topic is interesting and the study is timely. However, there are still some concerns to be addressed before publication.

1) The motivation of this study is not very clear, and the contributions of the proposed method should be highlighted.

2) The research background and significance were not clearly described. The research background should include an analysis of some current policies. Research significance should address the outcomes of this paper and point out improvements to be achieved in certain areas and judge the value of these research outcomes. It will be very important for the readers to judge whether the paper is useful for them to make a plan or obtain some suggestions.

3) A good review introduces the research status and the shortages of the current related studies, and find out how to solve the problem which has not been solved based on comprehensive literature review. The author introduced the relevant studies and conducted a summary; however, it is not sufficient for the readers to realize why the author choose the proposed method and what its innovations are. Please combine the logicality of the literature systematically to let the readers know why you choose this method.

4) The paper also lacks some latest relevant literature, so it is suggested to supplement them.

5) In Case Study Section, the author writes“The IC card and AV data of four days on October 1, 2, 3, and 4, 2019 were selected as the basic data of the example”; however, the reason why the author choose this period was not given. Authors can consider adding a paragraph to the discussion and introduce why they choose this period. Is the data extracted randomly?

6) In the Conclusions part, it is suggested to add limitations of the proposed model and to show future directions.

7) The English language throughout the paper should be polished carefully.

6. PLOS authors have the option to publish the peer review history of their article (what does this mean?). If published, this will include your full peer review and any attached files.

Reviewer #1: No

Reviewer #2: **Yes: **Chao Sun

Reviewer #3: No

---

## [Author Response · Author response to Decision Letter 0]

23 Aug 2021

Response to Reviewers

Reviewer #1:

This article develops a Kalman filter-based travel time prediction of the urban public transportation model. The reviewer has the following comments:

1. Please remove all comments and be professional!

Response: Thank you very much for your criticism and suggestions, which will make me more careful and professional in writing academic papers in the future. I apologize for my carelessness, unprofessionalism, and comments that should be deleted at the previous stage. I have revised all the formats and kept them standard before submitting them to you for review again.

2. In Section II, there is a syntax error in 'This literature review presents an overview of methods for predicting travel time published in the last five years is presented'.

Response: Thank you very much for your professionalism and carefulness in pointing out the spelling mistakes in the paper. All the mistakes have been corrected before submitting them to you for review again.

3. The flow in Section II would confuse readers a lot. Why not move 'Section II.B Kalman filtering' after the other four sub-sections, as this paper chooses Kalman filtering as the optimal algorithm? Present the pros and cons of SVM, GPS, PF, and NN, and then articulate the advantage of the Kalman filter over others.

Response: Thank you very much for your criticism and suggestions, which enables me to consider the structure of the whole paper more logically and clearly in future writing. I have adjusted the structure of the literature review. Indeed, the literature review on Kalman filtering is put at the end, which is more logical and clearer to readers.

4. The research gaps are still unclear. Sections I & II state why this paper chooses Kalman filter as the tool for travel time prediction, but none of the current research gaps have been presented.

Response: Thank you very much for your criticism and suggestions, the motivation and significance of selecting the Kalman filter as a travel time prediction tool have been added in Sections I & II.

5. The potential contributions and renovations of this paper are still unclear by the end of Section II. Please add potential contributions and renovations at the end of Section II.

Response: Thank you very much for your criticism and suggestions. In Section II, the significance and contribution of selecting the Kalman filter as a travel time prediction tool are added, and the renovations of this method are expounded.

6. The statement at the end of Section III.A.1), 'Given existing technical conditions, in-depth research on the prediction method of travel time of conventional transit (especially conventional transit under mixed traffic conditions) is of great theoretical significance and application value', is inconvincible, as this paper never mentions anything related to mixed traffic before or existing technical conditions.

Response: Thank you very much for your criticism and suggestions. Considering this paper's research scope and content, this part is reworded to ensure the logical integrity of the whole paper structure. The following paper will elaborate on the travel time prediction of urban public transport under mixed traffic conditions.

7. What is 'random factor' in Section III. A. 2). Please add definitions to terms before you use them.

Response: Thank you very much for your criticism and suggestions. 

A. Random factors include Road conditions (length, capacity, number of lanes, slope, grade, presence of bus lanes, type of median, and presence of crossings and the number of pedestrian crossing, roadside parking, fixed obstacles such as railway, intersection, control mode, etc.), traffic conditions, time, climate, traffic emergencies, vehicle conditions, drivers, passengers, etc.

B. Considering this paper's research scope and content, this part is reworded to ensure the logical integrity of the whole paper structure. Only a few random factors are briefly listed, and how detailed random factors affect travel time prediction will be illustrated in future studies.

8. The flow of this paper is very, very hard to follow. There is no connection between sub-sections and even paragraphs.

Response: Thank you very much for your criticism and suggestions. I rearranged the order of each part and the wording of some contents to ensure the whole paper's clear and complete logical structure.

9. The writing of this paper is very unfriendly to readers.

Response: Thank you very much for your criticism and suggestions. In addition to my own readjustment of the logical structure and wording of the text, I also asked professionals to help me polish the paper to ensure its readability.

10. Recommend authors to re-organize this paper's structure, improve the academic writing, and rephrase the whole paper before submission.

Response: Thank you very much for your criticism and suggestions, which helps me to pay more attention to my future academic paper writing. In addition to my own readjustment of the logical structure and wording of the text, I also asked professionals to help me polish the paper to ensure its readability.

Reviewer #2: 

This paper combines the Kalman filter model with the AVL and IC card intelligent bus technologies to predict urban public transport travel time. It is an interesting work. However, some fundamental problems need to be addressed:

1. The part of the Literature review lacks research contributions and the motivation of choosing the Kalman filter model.

Response: Thank you for your suggestions. The content is modified in the introduction, highlighting the motivation and contribution of selecting the Kalman filter as a travel time prediction tool.

2. The detailed parameters and processes of the neural network travel time prediction model should be explained since current comparisons are insufficient.

Response: The detailed parameters and process of the neural network travel time prediction model are not the focus of this paper. The comparison results between the neural network prediction model and the Kalman filter prediction model are described as evaluation parts. More explicit content will be illustrated in the following research content in the future.

3. Some data (such as bus code, pattern ID, departure time, et al.) has been detected with multiple devices (i.e., IC, AVL, and Bus dispatching). The authors should demonstrate the fusion process of these data.

Response: Thank you for your suggestions. 

A. In this paper, Madison's transit network data was provided by Wisconsin Traffic Operations and Safety Laboratory (these Figs are shown in S3, Figs 1 and 2). All data obtained for routes and stops are spatial data. This data was digitized and stored in a spatial database utilizing GIS technology for storage. 

B. The fusion process of multi-source data has a detailed process in the Supporting Files. Since data processing is not the focus of this paper, this part of the content will not be added to this paper.

4. The parts of Abstract and Conclusions should add the methodology and results with real engineering of the research.

Response: Thank you for your suggestions. The abstract and the conclusion are rewritten, and the methodology used in the paper and the research results of practical engineering significance are added.

5. Some minor writing mistakes "Kilman" should be "Kalman", "△" should be "Δ", not a triangle, et al.

Response: Thank you very much for your professionalism and carefulness in pointing out the spelling mistakes in the paper. All the mistakes have been corrected before submitting them to you for review again.

Reviewer #3:

Based on Automatic Vehicle Location (AVL) and IC data, this paper proposed various random traffic factors, constructed a transit travel time prediction model, and conducted a comparative analysis of the actual results. The topic is interesting, and the study is timely. However, there are still some concerns to be addressed before publication.

1. The motivation of this study is not very clear, and the contributions of the proposed method should be highlighted.

Response: Thank you for your suggestions. The content is modified in the introduction, highlighting the motivation and contribution of selecting the Kalman filter as a travel time prediction tool.

2. The research background and significance were not clearly described. The research background should include an analysis of some current policies. Research significance should address the outcomes of this paper and point out improvements to be achieved in certain areas and judge the value of these research outcomes. It will be essential for the readers to judge whether the paper helps make a plan or obtain some suggestions.

Response: Thank you very much for your criticism and suggestions, the background and significance of selecting the Kalman filter as a travel time prediction tool are expounded in Sections I & II.

3. A good review introduces the research status and the shortages of the current related studies and finds out how to solve the problem which has not been solved based on a comprehensive literature review. The author introduced the relevant studies and concluded; however, it is not sufficient for the readers to realize why the author chose the proposed method and its innovations. Please combine the logicality of the literature systematically to let the readers know why you choose this method.

Response: Thank you very much for your criticism and suggestions, the motivation and innovations of selecting the Kalman filter as a travel time prediction tool have been added in Sections I & II. The status of this research work is further clarified by comparing the advantages and disadvantages with other technologies.

4. The paper also lacks some latest relevant literature, so it is suggested to supplement them.

Response: Thanks to the suggestions, the reference part has been modified again, a few latest pieces of literature has been added, which makes up for the shortcomings of the review.

5. In Case Study Section, the author writes, "The IC card and AV data of four days on October 1, 2, 3, and 4, 2019 were selected as the basic data of the example"; however, the reason why the author chooses this period was not given. Authors can consider adding a paragraph to the discussion and introduce why they choose this period. Is the data extracted randomly?

Response: Thank you very much for your suggestions. 

A. In this paper, Madison's transit network data was provided by Wisconsin Traffic Operations and Safety Laboratory (these Figs are shown in S3, Figs 1 and 2). All data obtained for routes and stops are spatial data. This data was digitized and stored in a spatial database utilizing GIS technology for storage. 

B. The more recent data, the more convincing, but in 2020 and 2021, due to the impact of COVID-19, the public transport of Madison did not completely regular operation, so the data is not representative. Considering the stability of the urban public transport network and the climate characteristics of Madison, the data of October 1, 2, 3, and 4, 2019 was finally selected (the data of workdays are more representative). Different years of data did not affect the effectiveness of the travel time prediction model.

6. In the Conclusions part, it is suggested to add the proposed model's limitations and show future directions.

Response: Thank you very much for your criticism and suggestions. The limitations of this model have been mentioned, and future research and application directions have been added

7. The English language throughout the paper should be polished carefully.

Response: Thank you very much for your criticism and suggestions, which helps me to pay more attention to my future academic paper writing. In addition to my own readjustment of the logical structure and wording of the text, I also asked professionals to help me polish the paper to ensure its readability.

---

## [Decision Letter · Decision Letter 1]

22 Nov 2021

PONE-D-21-18697R1Travel Time Prediction of Urban Public Transportation Based on Detection of Single RoutesPLOS ONE

Dear Dr. Zhang,

Thank you for submitting your manuscript to PLOS ONE. After careful consideration, we feel that it has merit but does not fully meet PLOS ONE’s publication criteria as it currently stands. Therefore, we invite you to submit a revised version of the manuscript that addresses the points raised during the review process.

We look forward to receiving your revised manuscript.

Kind regards,

Jian Wang

Academic Editor

PLOS ONE

Journal Requirements:

Additional Editor Comments:

So far, I received the recommentations from three reviewers. Two recommended acceptance at the current form and one recommended major revision. After a careful check of this paper, I feel this paper has merits. Thereby, I made a decision of "minor revision". The authors should carefully address the comments from the third reviewer. Further, I found several typos in the abstract. The language of this paper should be improved. The authors are also encouraged to cite the following latest related studies.

Lu, L., He, Z., Wang, J.*, Chen, J., Wang, W. (2021). Estimation of lane-level travel time distributions under a connected environment. Journal of Intelligent Transportation Systems.

J. Huo, X. Fu, Z. Liu and Q. Zhang, Short-Term Estimation and Prediction of Pedestrian Density in Urban Hot Spots Based on Mobile Phone Data, in IEEE Transactions on Intelligent Transportation Systems, doi: 10.1109/TITS.2021.3096274.

Lu, L., Wang, J.*, Wu, Y., Chen, X. and Chan, C.Y., (2021). Real-time prediction model for vehicle individual travel time on an undersaturated signalized arterial. IEEE Intelligent Transportation Systems Magazine.

E. Chen, Z. Ye, C. Wang and M. Xu, Subway Passenger Flow Prediction for Special Events Using Smart Card Data, in IEEE Transactions on Intelligent Transportation Systems, vol. 21, no. 3, pp. 1109-1120, March 2020, doi: 10.1109/TITS.2019.2902405.

Lu, L., Wang, J., He, Z.*, Chan, C. Real-time estimation of freeway travel time with recurrent congestion based on sparse detector data, IET Intelligent Transport Systems.

Y. Liu, C. Lyu, X. Liu and Z. Liu, Automatic Feature Engineering for Bus Passenger Flow Prediction Based on Modular Convolutional Neural Network, in IEEE Transactions on Intelligent Transportation Systems, vol. 22, no. 4, pp. 2349-2358, April 2021, doi: 10.1109/TITS.2020.3004254.

Yang L A , Cheng L A , Zl A , et al. Exploring a large-scale multi-modal transportation recommendation system[J]. Transportation Research Part C: Emerging Technologies, 126.

H. Zhang, Y. Wu, H. Tan, H. Dong, F. Ding and B. Ran, Understanding and Modeling Urban Mobility Dynamics via Disentangled Representation Learning, in IEEE Transactions on Intelligent Transportation Systems, doi: 10.1109/TITS.2020.3030259.

Zhang B , Chen S , Ma Y , et al. Analysis on spatiotemporal urban mobility based on online car-hailing data[J]. Journal of Transport Geography, 2020, 82:102568.

Reviewers' comments:

Reviewer's Responses to Questions

**Comments to the Author**

1. If the authors have adequately addressed your comments raised in a previous round of review and you feel that this manuscript is now acceptable for publication, you may indicate that here to bypass the “Comments to the Author” section, enter your conflict of interest statement in the “Confidential to Editor” section, and submit your "Accept" recommendation.

Reviewer #1: (No Response)

Reviewer #2: All comments have been addressed

Reviewer #3: (No Response)

2. Is the manuscript technically sound, and do the data support the conclusions?

Reviewer #1: Partly

Reviewer #2: Yes

Reviewer #3: Yes

3. Has the statistical analysis been performed appropriately and rigorously? 

Reviewer #1: N/A

Reviewer #2: Yes

Reviewer #3: Yes

4. Have the authors made all data underlying the findings in their manuscript fully available?

Reviewer #1: (No Response)

Reviewer #2: Yes

Reviewer #3: Yes

5. Is the manuscript presented in an intelligible fashion and written in standard English?

Reviewer #1: No

Reviewer #2: Yes

Reviewer #3: Yes

6. Review Comments to the Author

Reviewer #1: The reviewer still has the following comments:

1) In the abstract, there is a syntax error in ‘Its greater time-variance and uncertainty make predictions for short travel times (≤35min) more subject to influence by random factors’. The ‘to influence’ should be ‘to be influenced’.

2) Syntax error. ‘It requires higher precision than makes long-term predictions and is more complicated.’

3) Syntax error. ‘The Kalman filter model has a higher precision in one-step-ahead prediction and can significantly automate massive data calculations to improve prediction accuracy.’

4) The writing of the abstract is hard for readers to understand the background and motivation of this study.

5) Syntax error. ‘it is essential to study the technology of travel time prediction to more fully meet the data characteristics of urban public transit and improve its applicability.’

6) The flow in Section II confuses readers a lot. In Section II. B, ‘While the models described in this section can solve the bus-to-station prediction problem to some degree, the influence factors these models considered are one-sided. SVM relies too much on kernel tricks to achieve predictions on a large scale. GPS overemphasizes the current state of the bus, degrading the prediction accuracy as the predicted distance increases. The PF considers only the time of the bus to the stop and ignores the spatial effect of the bus. The input used in the NN network is too one-sided and does not consider the comprehensive effect of time and space characteristics.’

Why does the summary of the four models occurs before the description of the SVM and PF? Please move to Section II.E.

7) Section II describes the advantage of the Kalman Filter over other four models in the travel time prediction. However, the research gaps and motivations on choosing the Kalman Filter to solve the problem in this study is still unclear. Please add more description about limitations of existing studies in travel time prediction with the Kalman Filter.

8) The research background, academic research gaps, motivations, potential real-world contributions, and renovations of this paper are still unclear by the end of Section II. Please well state the background, research gaps, motivations, and contributions in the Section I&II.

9) The English language and flow through this paper still need to be well organized.

Reviewer #2: All the comments are worked out by the authors. And this edition of manuscript is appropriate for publishing.

Reviewer #3: In this revised version of the manuscript authors have taken into account all the points in my previous comments. Therefore, in my opinion, this new version of the manuscript can be accepted for publication.

7. PLOS authors have the option to publish the peer review history of their article (what does this mean?). If published, this will include your full peer review and any attached files.

Reviewer #1: No

Reviewer #2: **Yes: **Chao Sun

Reviewer #3: No

---

## [Author Response · Author response to Decision Letter 1]

10 Dec 2021

Response to Reviewers

Reviewer #1:

The reviewer still has the following comments:

1. In the abstract, there is a syntax error in 'Its greater time-variance and uncertainty make predictions for short travel times (≤35min) more subject to influence by random factors'. The 'to influence' should be 'to be influenced'.

Response: Thank you very much for your professionalism and carefulness in pointing out the spelling mistakes in the paper. All the errors have been corrected before submitting them to you for review again.

2. Syntax error. 'It requires higher precision than makes long-term predictions and is more complicated.'

Response: Thank you very much for your professionalism and carefulness in pointing out the spelling mistakes in the paper. All the errors have been corrected before submitting them to you for review again.

3. Syntax error. 'The Kalman filter model has a higher precision in one-step-ahead prediction and can significantly automate massive data calculations to improve prediction accuracy.'

Response: Thank you very much for your professionalism and carefulness in pointing out the spelling mistakes in the paper. All the errors have been corrected before submitting them to you for review again.

4. The writing of the abstract is hard for readers to understand the background and motivation of this study.

Response: Thank you very much for your criticism and suggestions. I rewrote and adjusted that Section to ensure it was clear and complete.

5. Syntax error. 'it is essential to study the technology of travel time prediction to more fully meet the data characteristics of urban public transit and improve its applicability.'

Response: Thank you very much for your professionalism and carefulness in pointing out the spelling mistakes in the paper. All the errors have been corrected before submitting them to you for review again.

6. The flow in Section II confuses readers a lot. In Section II. B, 'While the models described in this Section can solve the bus-to-station prediction problem to some degree, the influence factors these models considered are one-sided. SVM relies too much on kernel tricks to achieve predictions on a large scale. GPS overemphasizes the current state of the bus, degrading the prediction accuracy as the predicted distance increases. The PF considers only the time of the bus to the stop and ignores the spatial effect of the bus. The input used in the NN network is too one-sided and does not consider the comprehensive effect of time and space characteristics.'

Response: Thank you for your suggestions. This content has been revised to ensure the integrity and clarity of this part of the content.

7. Why does the summary of the four models occur before describing the SVM and PF? Would you please move to Section II.E?

Response: Thank you for your suggestions. This content has been revised to ensure the integrity and clarity of this part of the content.

8. Section II describes the advantage of the Kalman Filter over the other four models in the travel time prediction. However, the research gaps and motivations on choosing the Kalman Filter to solve the problem in this study are still unclear. Would you please add more description about the limitations of existing studies in travel time prediction with the Kalman Filter?

Response: Thank you for your suggestions. The content of more description about limitations of existing studies in travel time prediction with the Kalman Filter has been revised

9. The research background, academic research gaps, motivations, potential real-world contributions, and renovations of this paper are still unclear by the end of Section II. Would you please state the background, research gaps, motivations, and contributions in Section I&II?

Response: Thank you for your suggestions. The content of background, research gaps, motivations, and contributions have been revised in the Section I&II

10. The English language and flow through this paper still need to be well organized.

Response: Thank you very much for your criticism and suggestions. In addition to my own readjustment of the logical structure and wording of the text, I also asked professionals to help me polish the paper to ensure its readability.

Reviewer #2: 

The authors work out all the comments. And this edition of the manuscript is appropriate for publishing.

Response: Thank you very much for your criticism and suggestions, which enables me to consider the structure of the whole paper more logically and clearly in future writing. Thank you very much for your reply and affirmation, which gives me the confidence to continue writing academic papers in the future. At the same time, based on the editor's advice, I adjusted the contents of the literature part and made the latest adjustments in grammar and diction.

Reviewer #3: 

In this revised version of the manuscript, the author has considered all the points in my previous comments. Therefore, in my opinion, this new version of the manuscript can be accepted for publication.

Response: Thank you very much for your criticism and suggestions, which enables me to consider the structure of the whole paper more logically and clearly in future writing. Thank you very much for your reply and affirmation, which gives me the confidence to continue writing academic papers in the future. At the same time, based on the editor's advice, I adjusted the contents of the literature part and made the latest adjustments in grammar and diction.

---

## [Decision Letter · Decision Letter 2]

28 Dec 2021

Travel Time Prediction of Urban Public Transportation Based on Detection of Single Routes

PONE-D-21-18697R2

Dear Dr. Zhang,

We’re pleased to inform you that your manuscript has been judged scientifically suitable for publication and will be formally accepted for publication once it meets all outstanding technical requirements.

Kind regards,

Jian Wang

Academic Editor

PLOS ONE

Additional Editor Comments (optional):

Reviewers' comments:

Reviewer's Responses to Questions

**Comments to the Author**

1. If the authors have adequately addressed your comments raised in a previous round of review and you feel that this manuscript is now acceptable for publication, you may indicate that here to bypass the “Comments to the Author” section, enter your conflict of interest statement in the “Confidential to Editor” section, and submit your "Accept" recommendation.

Reviewer #1: All comments have been addressed

2. Is the manuscript technically sound, and do the data support the conclusions?

Reviewer #1: Partly

3. Has the statistical analysis been performed appropriately and rigorously? 

Reviewer #1: (No Response)

4. Have the authors made all data underlying the findings in their manuscript fully available?

Reviewer #1: (No Response)

5. Is the manuscript presented in an intelligible fashion and written in standard English?

Reviewer #1: (No Response)

6. Review Comments to the Author

Reviewer #1: The latest manuscript has worked out all points in my previous comments. The reviewer still has the last comment:

1) In the second paragraph of Section I, please add references to support all statements about the research motivation. “Traditional travel time prediction methods based on statistical analysis or mathematical modeling are deficient in intelligence and have weak adaptability … it is necessary to study travel time prediction technology to more fully meet the data requirements in the operation analysis process of urban public transit and improve its applicability.”

7. PLOS authors have the option to publish the peer review history of their article (what does this mean?). If published, this will include your full peer review and any attached files.

Reviewer #1: No

---

## [Editor Report · Acceptance letter]

5 Jan 2022

PONE-D-21-18697R2 

Travel Time Prediction of Urban Public transportation Based on Detection of Single   Routes 

Dear Dr. Zhang:

I'm pleased to inform you that your manuscript has been deemed suitable for publication in PLOS ONE. Congratulations! Your manuscript is now with our production department. 

Kind regards, 

on behalf of

Dr. Jian Wang 

Academic Editor

PLOS ONE